# Ambient PM_2.5_ and PM_10_ Exposure and Respiratory Disease Hospitalization in Kandy, Sri Lanka

**DOI:** 10.3390/ijerph18189617

**Published:** 2021-09-12

**Authors:** Sajith Priyankara, Mahesh Senarathna, Rohan Jayaratne, Lidia Morawska, Sachith Abeysundara, Rohan Weerasooriya, Luke D. Knibbs, Shyamali C. Dharmage, Duminda Yasaratne, Gayan Bowatte

**Affiliations:** 1Department of Mathematics & Statistics, Texas Tech University, Lubbock, TX 79409, USA; tpriyank@ttu.edu; 2National Institute of Fundamental Studies, Hantana Road, Kandy 20000, Sri Lanka; mahesh.se@nifs.ac.lk (M.S.); rohan.we@nifs.ac.lk (R.W.); 3Postgraduate Institute of Science, University of Peradeniya, Peradeniya 20400, Sri Lanka; 4International Laboratory for Air Quality and Health, Queensland University of Technology, Brisbane, QLD 4000, Australia; r.jayaratne@qut.edu.au (R.J.); l.morawska@qut.edu.au (L.M.); 5Department of Statistics and Computer Science, Faculty of Science, University of Peradeniya, Peradeniya 20400, Sri Lanka; sachitha@pdn.ac.lk; 6National Center for Water Quality Research, University of Peradeniya, Peradeniya 20400, Sri Lanka; 7School of Public Health, The University of Sydney, Sydney, NSW 2006, Australia; luke.knibbs@sydney.edu.au; 8Allergy and Lung Health Unit, Melbourne School of Population and Global Health, University of Melbourne, Melbourne, VIC 3053, Australia; s.dharmage@unimelb.edu.au; 9Department of Medicine, Faculty of Medicine, University of Peradeniya, Peradeniya 20400, Sri Lanka; yasaratne@yahoo.com; 10Department of Basic Sciences, Faculty of Allied Health Sciences, University of Peradeniya, Peradeniya 20400, Sri Lanka

**Keywords:** ambient particulate matter, asthma, COPD, respiratory disease hospitalization, generalized additive model

## Abstract

Evidence of associations between exposure to ambient air pollution and health outcomes are sparse in the South Asian region due to limited air pollution exposure and quality health data. This study investigated the potential impacts of ambient particulate matter (PM) on respiratory disease hospitalization in Kandy, Sri Lanka for the year 2019. The Generalized Additive Model (GAM) was applied to estimate the short-term effect of ambient PM on respiratory disease hospitalization. As the second analysis, respiratory disease hospitalizations during two distinct air pollution periods were analyzed. Each 10 μg/m^3^ increase in same-day exposure to PM_2.5_ and PM_10_ was associated with an increased risk of respiratory disease hospitalization by 1.95% (0.25, 3.67) and 1.63% (0.16, 3.12), respectively. The effect of PM_2.5_ or PM_10_ on asthma hospitalizations were 4.67% (1.23, 8.23) and 4.04% (1.06, 7.11), respectively (*p* < 0.05). The 65+ years age group had a higher risk associated with PM_2.5_ and PM_10_ exposure and hospital admissions for all respiratory diseases on the same day (2.74% and 2.28%, respectively). Compared to the lower ambient air pollution period, higher increased hospital admissions were observed among those aged above 65 years, males, and COPD and pneumonia hospital admissions during the high ambient air pollution period. Active efforts are crucial to improve ambient air quality in this region to reduce the health effects.

## 1. Introduction

Respiratory diseases are among the major causes of death and disability globally. Approximately 65 million people suffer from chronic obstructive pulmonary disease (COPD), of whom 3 million die each year, making COPD the third leading cause of death worldwide [1]. Asthma is the most common chronic disease in children; globally, 334 million people suffer from asthma, posing a huge burden on families and national economies [1]. Air pollution has been identified as the leading environmental risk factor for many of these respiratory disorders, which can be prevented by taking necessary measures to reduce exposures [1,2]. However, in developing countries, the growing populations and increasing energy demand continuously lead to a further increase in ambient air pollution [3,4].

The association between ambient air pollution exposure and respiratory outcomes has been examined in many previous studies in terms of daily mortality, emergency department visits, and hospital admissions [5,6]. Among all studies conducted to date, those investigating the short-term effect of ambient air pollution on respiratory disease morbidity and mortality have been of particular interest given its ubiquitous nature of spread [5,7,8]. Air pollution can spread in a large geographical area within a short period and cause adverse health effects in exposed populations. Air pollution monitoring networks and well-documented hospital records are essential to evaluate the health effects of these short-term air pollution episodes. Therefore, such studies have mainly been from developed countries and only a limited number of studies were from developing countries [9]. For example, in Sri Lanka, few epidemiological studies have been targeted on air pollution’s health effects, and their main focus has been on indoor air pollution [10]. The nature and composition of ambient air pollution in developing countries such as Sri Lanka differ from those in developed countries. Therefore, the results of studies conducted in developed countries cannot be directly applied to developing country settings. This highlights the importance of local research to identify these associations.

Ambient air pollution is a growing problem in Sri Lanka and the neighboring countries, mainly due to the exponential growth in motor vehicles. In many developing countries, ambient air pollution is a neglected public health issue. It has been identified that the air quality in Kandy, the second largest city in Sri Lanka, is high compared to the capital Colombo [11]. This may be due to geographical characteristics as Kandy is centrally located in a small valley surrounded by high mountains. The city also has a high density of vehicles, narrow roads and frequent traffic congestion. Of note, increasing respiratory diseases such as COPD due to air pollution have been reported in Kandy [11]. According to a previous study measuring concentrations of ambient particulate matter (PM) < 10 µm in diameter (PM_10_) at twenty sites in the Kandy region, it was found that the mean 24-h PM_10_ concentration was 129 µg/m^3^ (range: 55–221 µg/m^3^), which is well above the World Health Organization (WHO) air quality guideline [12]. Our recent research also highlights that PM_2.5_ and PM_10_ concentrations in Kandy are high and exceed WHO guidelines [13].

The lack of evidence on the associations between exposure to ambient air pollution and health outcomes in the Kandy area is a significant knowledge gap that limits effective public health interventions. Therefore, in this study, we aimed to estimate the short-term effects of PM_2.5_ and PM_10_ on the risk of daily hospital admissions for respiratory diseases in Kandy area using the hospital admission data from the two major hospitals in 2019.

## 2. Materials and Methods

### 2.1. Study Population

The Kandy District in the Central Province of Sri Lanka has a land area of 1917 km^2^, with a total population of approximately 1.5 million. The City of Kandy is the capital of the province and it is situated in a low valley surrounded by mountain ranges. It has a resident population of approximately 120,000 [14]. The two main hospitals have a catchment of 18 out of 19 Divisional Secretariat regions in the Kandy District (Appendix A).

### 2.2. Air Pollution and Meteorology Data

Ambient air quality data (PM_2.5_ and PM_10_) were obtained from a validated small sensor unit established in a central location of Kandy city, Knowing Our Ambient Local Air-quality (KOALA). These KOALA monitors were developed by the International Laboratory for Air Quality and Health (ILAQH) at the Queensland University of Technology (QUT), Brisbane, Australia. This sensor continuously measured PM_2.5_ and PM_10_ using light scattering technology. Detailed information on sensor performance and validation has been published elsewhere [15]. Briefly, these sensors measure PM using a laser scattering principle to irradiate suspended particles in the air. The concentration of the particles is measured using the intensity of the scattered light spikes and the number of spikes, respectively [15,16]. The microcontroller in the sensor unit calculates and reports PM mass concentrations in three size ranges of 2.5 µm and 10.0 µm. Before deploying KOALA sensor units for field measurements, they were calibrated in the laboratory and under ambient conditions [17,18]. In this study, PM_2.5_ and PM_10_ measurements and meteorological measurements of temperature (°C) and relative humidity were obtained for the period 1 January 2019 to 31 December 2019.

### 2.3. Hospitalizations for Respiratory Diseases

Daily respiratory disease hospitalization data from two major hospitals in the Kandy area (Kandy National Hospital and Peradeniya Teaching Hospital) were obtained from 1 January 2019 to 31 December 2019. The data included date of admission, age, gender and the principal disease diagnosis coded according to the International Classification of Diseases, 10th revision (ICD-10). Hospitalization for respiratory diseases (ICD-10: J00–J99): pneumonia (ICD-10: J18), asthma ((ICD-10: 45) and chronic obstructive pulmonary disease (COPD, ICD-10: J40–J44 and J47) were included in the study. Further, we conducted an age- (≤64 years and 65+ years) and gender-stratified analysis for the association between PM exposure and respiratory disease hospital admissions.

### 2.4. Statistical Analysis

#### 2.4.1. Short Term Effects of PM_2.5_ and PM_10_ on Hospitalizations for Respiratory Diseases

A Generalized Additive Model (GAM) was applied to estimate the association between ambient air pollution (PM_2.5_, PM_10_) and respiratory diseases hospitalization [19,20]. We adopted the quasi-Poisson distribution in the GAM model [5,21]. To control the long-term trends of daily respiratory disease hospitalization, we used the cubic spline smoothing function of time [22]. Degrees of freedom of time was selected according to the minimum absolute values of the sum of partial autocorrelation function for lags up to 20 [5,22]. Representative variables for the weekends (DOW) and public holidays (holidays) were adjusted as dummy variables in the model. In addition, the natural smooth functions of average temperature and humidity were included in the model as confounders on the association between air pollution and respiratory disease hospitalization. The 14-day moving average of mean temperature and the humidity of the current day used in this study were based on a previous study with a similar analytical approach [5]:Log [E(Yt)] = α + s (time, df) + s (temperature, df) + s (humidity, df) + DOW+ holiday+ βZt 

E(Yt) represents the expected number of respiratory disease hospitalizations on day t; α denotes the intercept; S is the spline smoothing function for the variables of time, temperature and humidity; df is the degrees of freedom; β denotes the coefficient for air pollutants; and Zt denotes the concentration of air pollutant on day t. To investigate the delayed effects of air pollutants on respiratory diseases hospitalization, we used day lags of PM_2.5_ and PM_10_ (e.g., lag0, lag1, lag2, lag3, lag4, lag5).

#### 2.4.2. Respiratory Disease Hospitalizations in Two Distinct Air Pollution Seasons in 2019

We also identified two distinct seasons of ambient air pollution in 2019, based on the measured average PM_2.5_. The high air pollution period was defined as when the daily average PM_2.5_ levels were greater than the WHO guideline (25 µg/m^3^). In contrast, the low air pollution period was defined as the period where the low levels of average PM_2.5_ were below the WHO guideline. We modeled the associations between respiratory disease hospitalization in the high pollution period by selecting three representative months for both high (1 March 2019 to 31 May 2019) and low ambient air pollution (1 August 2019 to 31 October 2019) and compared the periods using the low pollution period as the reference. Assuming that the change in the baseline population within six months is negligible, we used rate ratios (RRs), defined as the ratio of the number of hospitalizations in the high ambient air pollution period and the reference period. Then, 95% confidence intervals were obtained using the following formula, where a and b are the number of hospitalizations in the high and low periods [23]:(1)$$\exp\left(\ln\;\mathrm{RR}\pm 1.96\sqrt{\frac{1}{a}+\frac{1}{b}}\right)$$

## 3. Results

The locations of the air monitoring site and two hospitals are shown in Figure 1. We included 9709 respiratory disease hospitalizations from 1 January 2019 to 31 December 2019, of which 53.77% were male and 46.23% were female (Appendix A). Table 1 provides descriptive statistics for daily respiratory disease hospitalizations, meteorological factors and air pollutants. Daily mean hospitalization cases were 30.63 for total respiratory diseases, including 1.51, 7.13 and 6.35 for pneumonia, COPD and asthma, respectively. For air pollutants, the daily average concentrations for PM_2.5_ and PM_10_ were 34.48 µg/m^3^ and 38.52 µg/m^3^, respectively. The average daily temperature was 27.65 °C, and the average relative humidity was 73.06%.

### 3.1. Associations between Short Term PM_2.5_ and PM_10_ Exposure and Hospitalizations for Respiratory Diseases

Results of the PM_2.5_ and PM_10_ pollutant models at different lags are presented in Table 2. After adjusting for the confounding factors, PM_2.5_ and PM_10_ were associated with all respiratory disease hospitalizations (*p* < 0.05). Associations of a 10 µg/m^3^ increment in PM_2.5_ and percent changes of all respiratory disease hospitalizations were found at lag0 with 1.95% (95%CI: 0.25, 3.67), PM_10_ at lag0 with 1.63% (95%CI: 0.16, 3.12). In this analysis, we have investigated lag effects for more than a month for both pollutants. However, significant associations were found at lag0 and lag1. Therefore, we have presented the results only up to lag5.

The lag effects of the percent increase in hospitalization for different types of respiratory diseases with a 10 µg/m^3^ increase in pollutants are reported in Table 2. In gender-stratified analysis, at lag0, associations were found for the estimated effects of PM_2.5_ and PM_10_ in males. At lag0, a 10 µg/m^3^ increase in PM_2.5_ exposure was associated with increased respiratory disease hospitalization in both males and females (2.24%, 95%CI: 0.08, 4.44; 2.35%, 95%CI: 0.08, 4.68, respectively). The 65+ years age group had a higher risk associated with both PM_2.5_ and PM_10_ exposure and hospital admissions for all respiratory diseases (2.74% and 2.28% at lag0, respectively). Estimated effects of PM_2.5_ and PM_10_ exposure on ≤65 years group were not significant at lag0. However, both pollutants’ 10 µg/m^3^ increase showed associations at lag1, with increased respiratory diseases of 2.23% and 1.93%. For both pollutants, the largest percent increase values occurred at the current day 10 µg/m^3^ increase in PM_2.5_ or PM_10_ on asthma hospitalizations (4.67% and 4.04%, respectively) (*p* < 0.05). The association between increased concentration of PM_2.5_ and PM_10_ and hospitalization for pneumonia and COPD was not statistically significant at any of the lags (Table 2).

### 3.2. Increased Respiratory Disease Hospitalizations in High Air Pollution Period in 2019

As shown in Figure 2, in 2019 two distinct periods of air pollution were identified. The high air pollution period ranged from January to May 2019 and the low air pollution period from June to December 2019. In the high pollution period, the mean daily levels of PM_2.5_ (48.77 µg/m^3^, SD = 14.87) and PM_10_ (54.97 µg/m^3^, SD = 17.13) were approximately two times higher compared to the low air pollution period. For our analysis, we selected three representative months of high air pollution (1 March 2019 to 31 May 2019) and three representative months of low air pollution (1 August 2019 to 31 October 2019) as the reference period. The number of respiratory disease cases referred to the two hospitals were 2427 and 2946 during low and high air pollution periods, respectively. The gender distribution of hospitalizations for the period was 23.49% and 30.30% for males during low and high air pollution periods, respectively, and 21.68% and 24.53% for females during low and high air pollution periods, respectively. Notably, 9.77% and 13.16% of respiratory disease hospitalization were due to COPD during the low and high air pollution periods, respectively (Appendix A).

There was a significantly higher average number of daily hospital admissions in the high air pollution period for total respiratory diseases (32.73, SD = 9.07, *p* < 0.0001), male respiratory diseases (18.09, SD = 5.15, *p* < 0.0001), female respiratory diseases (14.64, SD = 5.81, *p* = 0.07), both age groups (≤65 years: 20.51, SD = 6.69, *p* = 0.004; >65 years: 12.22, SD = 3.96, *p* < 0.0001) and COPD (7.86, SD = 3.26, *p* < 0.0001) (Table 3).

Compared to the low air pollution reference period, increased hospital admissions were observed for all respiratory diseases during the high air pollution period (RR = 1.21 (95% CI 1.15,1.28)). During the high air pollution period, increased respiratory disease hospital admissions were observed among elders (65+ years) (RR = 1.31 (95%CI 1.20,1.43)) compared to ≤ 65 year olds (RR = 1.16 (95%CI 1.09,1.24)), compared with the low air pollution period. Similar associations were observed for COPD (RR = 1.35 (95% CI 1.20,1.51)) and pneumonia (RR = 1.58 (95%CI 1.13,2.20)) during the high air pollution period compared with the low air pollution period (Table 4). In addition, respiratory disease hospital admissions were higher for males (RR = 1.29 (95%CI 1.20,1.39)) during the high pollution period compared with the low air pollution period.

## 4. Discussion

The present study showed significant associations between PM_2.5_ and PM_10_ levels and increased risk of respiratory disease hospital admissions. A 10 µg/m^3^ increase in PM_2.5_ and PM_10_ were associated with an increased risk of respiratory disease hospitalization by up to 1.95 % and 1.63%, respectively, on the same day. Moreover, those aged above 65 were more vulnerable to PM_2.5_ and PM_10_ exposure. Furthermore, our analysis reported significant effects of short term PM_2.5_ and/or PM_10_ exposure on asthma hospitalizations. In our analysis comparing respiratory disease hospital admissions in high and low air pollution periods, we identified a statistically significant increase in hospital admissions for all respiratory diseases, in males, and for COPD and pneumonia during the high air pollution period.

Our findings showed that a 10 µg/m^3^ increment in PM_2.5_ on the current day was associated with a 2.0% increase in overall respiratory disease hospitalization, which is in line with most previous studies. A study in Eastern China found that an increase of 10 µg/m^3^ in PM_2.5_ corresponded to a 1.4% (95% CI: 0.7,2.1) increase in respiratory emergency room visits in urban areas, and a 1.5% (95%CI: 0.4,2.6) rise for the suburban population. In the same study, short term PM_10_ exposure was linked with an increase of 1.63% at lag0 for all respiratory disease hospital admissions per 10μg/m^3^ increase in PM_10_ concentration [24]. A study from Palermo, Italy found that a 10μg/m^3^ increase of PM_10_ was associated with a 2.20% rise in hospital admissions for respiratory diseases [25].

In our gender-stratified analysis, significant effects of PM_2.5_ on hospital admission at lag0 were observed in both females and males (2.35% (95% CI: 0.08–4.68 %) and 2.24% (95% CI: 0.08–4.44%), respectively). Older people had higher respiratory disease hospital admissions, with percent changes of 2.74% and 2.28% in those aged above 65 years at lag0 due to a 10 μg/m^3^ rise of PM_2.5_ and PM_10_, respectively. Many of the previous studies have identified older adults as a high-risk group for air pollution exposure [26]. For a 10 µg/m^3^ increment in PM_2.5_ and PM_10_ on the previous day, respiratory disease hospitalization in those aged less than 65 years also showed statistically significant increases of 2.23% and 1.93%, respectively, while for the current-day exposure, results were statistically insignificant. In line with previous studies, we observed a strong significant association between short-term PM_2.5_ and/or PM_10_ exposure and asthma hospitalization [27,28]. Percent changes of hospitalization due to pneumonia and COPD were not significant for either pollutant. This non-significance attributed to pneumonia hospitalization was consistent with a previous analysis [29]. However, a meta-analysis of 12 studies to evaluate the association between PM_2.5_ exposure and COPD hospitalization found that a 10-µg/m^3^ increase in PM_2.5_ at lag 0–7 days could lead to a 3.1% (1.6–4.6%) increase in COPD hospitalization [30].

During the high air pollution period a high-rate ratio was observed for male respiratory disease hospitalization compared to female respiratory disease hospitalization. We can speculate the reason for this finding might be males spending more time outside during the high air pollution period (and thus having a substantially increased exposure time to air pollutants) because of their employment and social factors [31]. Our age-stratified analysis showed that older people (aged above 65 years) were more sensitive to ambient air pollution, which is consistent with most of the previous studies [32,33]. This finding is quite predictable because of decreased lung function and poorer immune system function in older age groups [34,35].

The two distinct air pollution periods observed in Kandy were demarcated by patterns of rainfall, with high rainfall generally starting after April [36]. Regional air pollution significantly influences Sri Lanka’s air pollution profile, where pollution from the Indian subcontinent drives down towards the country with the wind and contributes to the island’s high air pollution, especially during the dry months [37]. Compared to the low air pollution reference period, we identified a statistically significant increase in hospital admissions for all respiratory diseases in both genders and age groups and COPD and pneumonia during the high air pollution period. This highlights that high air pollution levels during Kandy’s dry season significantly impact hospital admissions for respiratory diseases. A similar analysis in Beijing which examined the association between air quality and respiratory disease hospital visits in a Beijing hospital during a comparable high-smog period revealed rate ratios of 1.74 for emergency room visits and 1.16 for outpatient visits for respiratory diseases [23].

There are some limitations in our analysis. PM_2.5_ and PM_10_ data and meteorological factors in this study were obtained from one fixed air quality monitoring sensor unit in Kandy, which may not represent the population’s total exposure. Therefore, our study might underestimate the effects of PM_2.5_ and PM_10_ on respiratory disease hospitalization. However, we analyzed air pollution data from the network of four units of KOALA, covering a large part of Kandy, that was taken from February to March in 2020. According to the average value of the daily mean level of PM_2.5_ and PM_10_ and their lower standard deviations of each of these four KOALA units, the pollution was generally uniformly distributed around the city. Therefore, we are confident that this result may be applied to the actual period of monitoring and that we can safely use the results from our monitoring to represent the average concentrations in the Kandy region. (See Appendix A). Moreover, from 22 June to 1 August 2019, the air pollution data were not available due to a technical issue in the sensor. Our estimates on air pollutants’ effects on pneumonia were obtained using a relatively low number of hospital admissions. Therefore, further studies are crucial to confirm the relationship between ambient air pollution and specific respiratory diseases in Kandy using both air pollution and hospital admissions data over more extended periods. Only the main exposure of interest is considered in these observational studies, and other important confounding factors are not included due to the unavailability of such data.

In 2018, the majority of hourly average PM_10_ and PM_2.5_ measurements coming from two monitoring stations (one in Colombo and the other in Kandy) exceeded the Sri Lankan standards. More significantly, the measurements from Kandy are often found to be higher than the corresponding values of Colombo [11]. This highlights the significant need and urgency to estimate the health effects of air pollution in the Kandy City area.

## 5. Conclusions

Our investigations of the short-term effects of PM_2.5_ and PM_10_ on respiratory disease hospitalization identified significant positive associations between PM_2.5_ and PM_10_ concentration at different lag days and hospitalization for respiratory diseases. Higher hospital admissions for respiratory diseases were observed during the high ambient air pollution period compared with the low ambient air pollution period. Our analysis suggests that males and older people are more vulnerable to respiratory disease hospitalization during high ambient air pollution period. To our knowledge, this is the first study to estimate PM and respiratory hospital admissions using continuous air pollution measurements in Sri Lanka. Overall, in the region, air pollution monitoring and quality hospital admission data are limited. Therefore, this study provides evidence that applications of low-cost air pollution measurements to investigate health risks associated with PM air pollution can be adopted by such low-resource settings. Furthermore, our findings highlight the importance of regular widespread air pollution monitoring and identify the need for immediate actions and regulatory measures to reduce PM air pollution levels, which will in turn lead to reduced respiratory disease hospital admissions.

## Figures and Tables

**Figure 1 ijerph-18-09617-f001:**
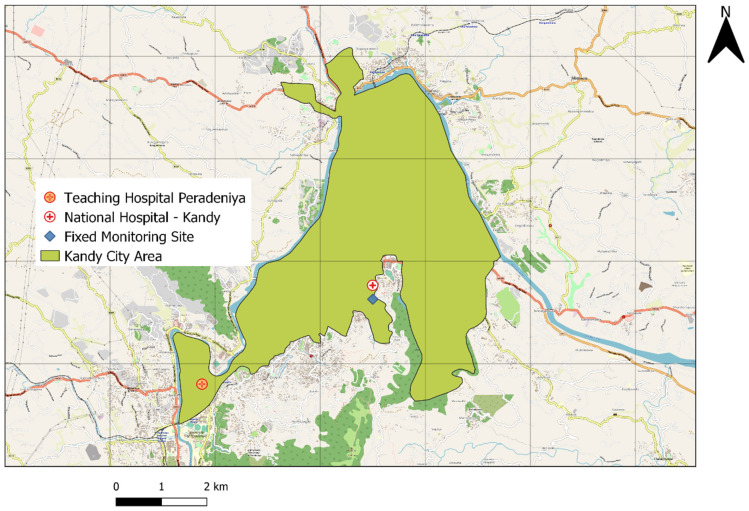
The locations of the air monitoring site and hospitals in Kandy.

**Figure 2 ijerph-18-09617-f002:**
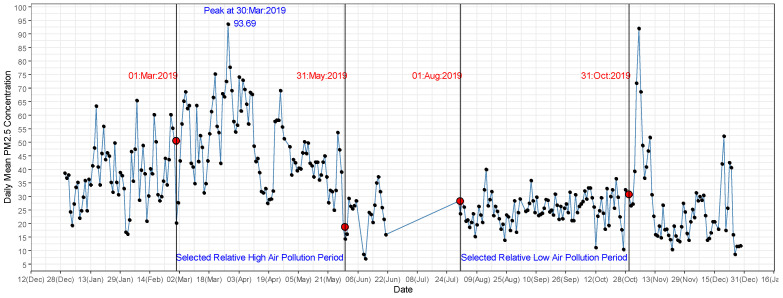
Variation of average daily PM_2.5_ (μg/m^3^) over the period 1 January 2019 to 31 December 2019 in Kandy.

**Table 1 ijerph-18-09617-t001:** Descriptive statistics for daily respiratory disease hospitalization, daily meteorological factors and air pollutants in Kandy, 2019.

Factors	Mean (SD)	Min	P25	Median	P75	Max
Meteorological factors (daily mean)						
Temperature °C	27.65 (1.99)	22.21	26.33	27.59	28.85	40.03
Humidity %	73.06 (8.56)	48.80	66.30	72.52	79.04	97.32
Air pollutants (µg/m^3^)						
PM_2.5_	34.48 (15.75)	7.02	23.54	30.65	42.73	93.69
PM_10_	38.52 (18.23)	7.53	25.86	33.90	47.93	107.65
Respiratory disease (mean daily hospital admissions)						
All	30.63 (9.84)	2	24	29	37	60
Male	16.52 (5.79)	0	13	16	20	36
Female	14.16 (5.47)	1	11	14	17	32
≤64 years	19.33 (6.88)	2	15	19	23	43
65+ years	11.33 (4.43)	0	8	11	14	27
Pneumonia	1.51 (0.74)	0	1	1	2	4
COPD	7.13 (3.25)	0	5	7	9	20
Asthma	6.35(2.99)	0	4	6	8	16

**Table 2 ijerph-18-09617-t002:** The percent change for respiratory disease hospitalizations associated with short-term 10 µg/m^3^ increases in particulate matter air pollutant concentrations.

Hospitalization Due to All Respiratory Diseases (%)				
	PM_2.5_		PM_10_	
Lags	PM_2.5_	95% CI	PM_10_	95% CI
Lag0	1.95 *	0.25, 3.67	1.63 *	0.16, 3.12
Lag1	1.31	−0.40, 3.06	1.12	−0.37, 2.63
Lag2	−0.07	−1.81, 1.69	−0.06	−1.57,1.47
Lag3	−0.64	−2.38, 1.13	−0.57	−2.09, 0.96
Lag4	−0.29	−2.06, 1.52	−0.25	−1.79, 1.32
Lag5	−0.55	−2.33, 1.27	−0.50	−2.05, 1.08
Hospitalization due to respiratory diseases—males only (%)				
Lags	PM_2.5_	95% CI	PM_10_	95% CI
Lag0	2.24 *	0.08, 4.44	1.96 *	0.09, 3.87
Lag1	1.62	−0.55, 3.84	1.42	−0.47, 3.33
Lag2	0.70	−1.50, 2.94	0.64	−1.27, 2.59
Lag3	−0.24	−2.44, 2.00	−0.20	−2.11, 1.75
Lag4	0.08	−2.14, 2.36	0.14	−1.80, 2.12
Lag5	0.30	−1.94, 2.59	0.26	−1.69, 2.25
Hospitalization due to respiratory diseases—female only (%)				
Lag0	2.35 *	0.08, 4.68	1.89	−0.09, 3.90
Lag1	1.71	−0.60, 4.08	1.44	−0.57, 3.48
Lag2	−0.19	−2.53, 2.20	−0.20	−2.23, 1.88
Lag3	−0.33	−2.69, 2.08	−0.33	−2.38, 1.77
Lag4	0.08	−2.32, 2.53	0.00	−2.09, 2.14
Lag5	−0.45	−2.86, 2.01	−0.45	−2.54, 1.69
Respiratory diseases hospitalization of age 65+ group (%)				
Lag0	2.74 *	0.29, 5.26	2.28 *	0.15, 4.46
Lag1	0.27	−2.21, 2.82	0.19	−1.97, 2.40
Lag2	−1.08	−3.59, 1.49	−0.95	−3.14, 1.29
Lag3	−0.62	−3.15, 1.99	−0.48	−2.70, 1.78
Lag4	−0.87	−3.43, 1.77	−0.70	−2.94, 1.59
Lag5	0.01	−2.58, 2.66	0.04	−2.21, 2.35
Respiratory diseases hospitalization of age ≤65 group (%)				
Lag0	1.76	−0.31, 3.87	1.49	−0.30, 3.32
Lag1	2.23 *	0.14, 4.37	1.93 *	0.11, 3.78
Lag2	0.82	−1.30, 2.99	0.73	−1.12, 2.61
Lag3	−0.39	−2.50, 1.77	−0.39	−2.23, 1.49
Lag4	0.33	−1.83, 2.53	0.27	−1.61, 2.18
Lag5	−0.45	−2.62, 1.77	−0.45	−2.34, 1.48
Hospitalization due to COPD (%)				
Lag0	2.96	−0.28, 6.30	2.42	−0.39, 5.31
Lag1	1.69	−1.58, 5.07	1.33	−1.52, 4.25
Lag2	−1.39	−4.66, 2.00	−1.27	−4.13, 1.68
Lag3	0.37	−2.96, 3.82	0.17	−2.73, 3.16
Lag4	0.14	−3.22, 3.63	0.06	−2.88, 3.08
Lag5	2.54	−0.86, 6.06	2.19	−0.76, 5.23
Hospitalization due to pneumonia (%)				
Lag0	0.37	−4.35, 5.32	0.38	−3.71, 4.65
Lag1	0.03	−4.71, 5.00	0.02	−4.09, 4.30
Lag2	−0.58	−5.37, 4.47	−0.29	−4.46, 4.06
Lag3	1.73	−3.23, 6.94	1.57	−2.71, 6.04
Lag4	0.64	−4.49, 6.05	0.42	−4.01, 5.07
Lag5	2.67	−2.44, 8.04	2.01	−2.40, 6.61
Hospitalization due to asthma (%)				
Lag0	4.67 *	1.23, 8.23	4.04 *	1.06, 7.11
Lag1	3.27	−0.24, 6.90	2.85	−0.19, 5.99
Lag2	2.62	−0.95, 6.32	2.26	−0.84, 5.46
Lag3	−0.11	−3.67, 3.59	−0.02	−3.13, 3.19
Lag4	−0.84	−4.39, 2.85	−0.79	−3.89, 2.42
Lag5	−0.07	−3.65, 3.65	−0.24	−3.37, 2.99

* *p* < 0.05.

**Table 3 ijerph-18-09617-t003:** Descriptive statistics of daily air pollution levels and daily respiratory disease hospital admissions in two air pollution periods (in selected months).

	Air Pollution Period	Minimum	Mean (SD)	Maximum	*p* Value *
PM_2.5_ (µg/m^3^)	Low	10.48	25.31 (5.38)	40.04	<0.0001
	High	18.73	48.77 (14.87)	93.69	
PM_10_ (µg/m^3^)	Low	11.28	27.74 (5.94)	44.49	<0.0001
	High	20.81	54.97 (17.13)	107.46	
Temperature (°C)	Low	22.21	26.36 (1.64)	29.98	<0.0001
	High	24.80	28.66 (1.33)	30.84	
Humidity (%)	Low	64.88	80.07 (6.80)	97.32	<0.0001
	High	48.80	68.20 (7.98)	85.51	
All	Low	12	26.97 (7.14)	46	<0.0001
	High	13	32.73 (9.07)	54	
Male	Low	3	14.02 (4.88)	31	<0.0001
	High	9	18.09 (5.15)	36	
Female	Low	3	12.94 (3.86)	26	0.07
	High	2	14.64 (5.81)	31	
≤64 years	Low	7	17.62 (5.16)	37	0.004
	High	9	20.51 (6.69)	38	
65+ years	Low	2	9.34 (3.82)	20	<0.0001
	High	4	12.22 (3.96)	24	
Pneumonia	Low	0	1.53 (0.69)	3	0.6561
	High	0	1.61(0.76)	4	
COPD	Low	0	5.90 (2.75)	14	<0.0001
	High	2	7.86 (3.26)	17	
Asthma	Low	1	5.93 (2.52)	15	0.5948
	High	0	6.28 (2.93)	15	

* Welch’s test results.

**Table 4 ijerph-18-09617-t004:** Associations of respiratory disease hospital admissions during the high air pollution period compared to the low air pollution period.

Respiratory Disease Hospitalizations	Rate Ratio (95% CI)
All	1.21 (1.15, 1.28)
Male	1.29 (1.20, 1.39)
Female	1.13 (1.05, 1.22)
65+ years	1.31 (1.20, 1.43)
≤64 years	1.16 (1.09, 1.25)
COPD	1.35 (1.20, 1.51)
Pneumonia	1.58 (1.13, 2.20)
Asthma	1.05 (0.93, 1.18)

## Data Availability

The data presented in this study are available on request from the corresponding author.

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
