# Peer review of "Ambient PM2.5 and PM10 Exposure and Respiratory Disease Hospitalization in Kandy, Sri Lanka"

_ijerph, 2021, doi:10.3390/ijerph18189617_

Round 1
Reviewer 1 Report
This ms. informs about the effect of ambient PM2.5 and PM10 exposure and respiratory disease hospitalization in Kandy, Sri Lanka. The level of PM2.5 34.5±15 7 mg/m3 is very high. Authors evaluated 9 709 respiratory diseases hospitalization during all year 2019. They very carefully analyzed the effect of short term 10 mg/m3 increases. The most significant effect was observed in the hospitalization due to asthma. When they compared the effect of PM2.5 low exposure (25.3±5.4 mg/m3) vs. high exposure (48.8±14.9 mg/m3), they observed the effect of high exposure on hospitalization, more pronounced in males, aged above 65 years, COPD and pneumonia.
Question: authors mentioned in conclusions l. 345 that females are more vulnerable to respiratory diseases hospitalization, they should checked data in Table 3 and Table 4 males vs. females.
They could also emphasize the necessity to decrease ambient PM2.5 concentrations in Sri Lanka to decrease its impact to human health.
Author Response
We thank the reviewers for providing suggestions to improve the manuscript.Reviewer 1:
This ms. informs about the effect of ambient PM2.5 and PM10 exposure and respiratory disease hospitalization in Kandy, Sri Lanka. The level of PM2.5 34.5±15 7 mg/m3 is very high. Authors evaluated 9 709 respiratory diseases hospitalization during all year 2019. They very carefully analyzed the effect of short term 10 mg/m3 increases. The most significant effect was observed in the hospitalization due to asthma. When they compared the effect of PM2.5 low exposure (25.3±5.4 mg/m3) vs. high exposure (48.8±14.9 mg/m3), they observed the effect of high exposure on hospitalization, more pronounced in males, aged above 65 years, COPD and pneumonia.
Question: authors mentioned in conclusions l. 345 that females are more vulnerable to respiratory diseases hospitalization, they should check data in Table 3 and Table 4 males vs. females.
We thank the reviewer for pointing out this. We have now corrected the conclusion.
Line 429-431: “Our analysis suggests that males and older people are more vulnerable to respiratory diseases hospitalization during high ambient air pollution period.”
They could also emphasize the necessity to decrease ambient PM2.5 concentrations in Sri Lanka to decrease its impact to human health.
Thank you. We have now updated the conclusion by considering suggestions of reviewer 1 and reviewer 2.
Line 433-439: “Overall, in the region, air pollution monitoring and quality hospital admission data are limited. Therefore, this study provides evidence that applications of low-cost air pollution measurements to investigate health risks associated with PM air pollution can be adopted by such low-resource settings. Furthermore, our findings highlight the importance of regular widespread air pollution monitoring and identify the need for immediate actions and regulatory measures to reduce PM air pollution levels, which will intern lead to reduce respiratory disease hospital admissions.”
Reviewer 2 Report
Evidence of associations between exposure to ambient air pollution and health outcomes17 are sparse in the South Asian region due to limited air pollution exposure and quality heath-data. Accordingly, the impact of environment on health is becoming more and more serious and obvious, so it is very worth studying. Taking Sri Lanka as an example, this paper makes an empirical analysis and draws an enlightening conclusion, which is worthy of affirmation.
1. Why in Table used Lag 5? please explain more in details.
2. The Conclusions section should include some content which is related to policy discussion. In the final conclusion, it is a little hasty and insufficient. Here should provide the good experience of this study that can be extended to other regions in the world, which is an important enlightenment of environmental governance. Therefore, this part should analyze the policy implicatioon in three parts.
3. The sentences in the article should be more concise and clear. Some statements in this study should be supported by more literature. Since the air pollution is highly related to Energy cnsumption, so some fresh Energy paper can be added as reeferences, eg:
Sun, H., Kporsu, A.K, Taghizadeh-Hesary, F., Edziah, B.K., 2020. Estimating environmental efficiency and convergence: 1980 to 2016. Energy 118224.
Author Response
We thank the reviewer for providing suggestions to improve the manuscript.Reviewer 2:
Evidence of associations between exposure to ambient air pollution and health outcomes17 are sparse in the South Asian region due to limited air pollution exposure and quality heath-data. Accordingly, the impact of environment on health is becoming more and more serious and obvious, so it is very worth studying. Taking Sri Lanka as an example, this paper makes an empirical analysis and draws an enlightening conclusion, which is worthy of affirmation.
- Why in Table used Lag 5? please explain more in details.
We thank the reviewer for constructive comments and suggestions.
In our analysis, we have conducted the analysis up to a lag period of more than a month. However, we did not find any significant associations in these extended lag periods. Therefore, to highlight the importance of short-term ambient air pollution and its effect on respiratory hospital admissions, the results presented were restricted to a lag period of 5 days.
Line 230-233: “In this analysis, we have investigated lag effects more than a month for both pollutants. However, significant associations were found in lag 0 and lag 1. Therefore, we have presented the results only up to lag5.”
- The Conclusions section should include some content which is related to policy discussion. In the final conclusion, it is a little hasty and insufficient. Here should provide the good experience of this study that can be extended to other regions in the world, which is an important enlightenment of environmental governance. Therefore, this part should analyze the policy implication in three parts.
We have now updated the conclusion taking the reviewer’s suggestions.
Line 433-439: “Overall, in the region, air pollution monitoring and quality hospital admission data are limited. Therefore, this study provides evidence that applications of low-cost air pollution measurements to investigate health risks associated with PM air pollution can be adopted by such low-resource settings. Furthermore, our findings highlight the importance of regular widespread air pollution monitoring and identify the need for immediate actions and regulatory measures to reduce PM air pollution levels, which will intern lead to reduce respiratory disease hospital admissions.”
- The sentences in the article should be more concise and clear. Some statements in this study should be supported by more literature. Since the air pollution is highly related to Energy cnsumption, so some fresh Energy paper can be added as reeferences, eg:
Sun, H., Kporsu, A.K, Taghizadeh-Hesary, F., Edziah, B.K., 2020. Estimating environmental efficiency and convergence: 1980 to 2016. Energy 118224.
As suggested by the reviewer we have updated some sentences with supporting literature.
We have added the suggested reference to the manuscript.
Line 43-45: “However, in developing countries, the growing populations and increasing energy demand continuously lead to a further increase in ambient air pollution”